# CLOCS: Contrastive Learning of Cardiac Signals Across Space, Time, and Patients

## Abstract

The healthcare industry generates troves of unlabelled physiological data. This data can be exploited via contrastive learning, a self-supervised pre-training method that encourages representations of instances to be similar to one another. We propose a family of contrastive learning methods, CLOCS, that encourages representations across space, time, *and* patients to be similar to one another. We show that CLOCS consistently outperforms the state-of-the-art methods, BYOL and SimCLR, when performing a linear evaluation of, and fine-tuning on, downstream tasks. We also show that CLOCS achieves strong generalization performance with only 25% of labelled training data. Furthermore, our training procedure naturally generates patient-specific representations that can be used to quantify patient-similarity.

## 1 Introduction

At present, the healthcare system is unable to sufficiently leverage the large, unlabelled datasets that it generates on a daily basis. This is partially due to the dependence of deep learning algorithms on high quality labels for good generalization performance. However, arriving at such high quality labels in a clinical setting where physicians are squeezed for time and attention is increasingly difficult. To overcome such an obstacle, self-supervised techniques have emerged as promising methods. These methods exploit the unlabelled dataset to formulate pretext tasks such as predicting the rotation of images (Gidaris et al., 2018), their corresponding colourmap (Larsson et al., 2017), and the arrow of time (Wei et al., 2018). More recently, contrastive learning was introduced as a way to learn representations of instances that share some context. By capturing this high-level shared context (e.g., medical diagnosis), representations become invariant to the differences (e.g., input modalities) between the instances.

Contrastive learning can be characterized by three main components: 1) a positive and negative set of examples, 2) a set of transformation operators, and 3) a variant of the noise contrastive estimation loss. Most research in this domain has focused on curating a positive set of examples by exploiting data temporality (Oord et al., 2018), data augmentations (Chen et al., 2020), and multiple views of the same data instance (Tian et al., 2019). These methods are predominantly catered to the image-domain and central to their implementation is the notion that shared context arises from the same instance. We believe this precludes their applicability to the medical domain where physiological time-series are plentiful. Moreover, their interpretation of shared context is limited to data from a common source where that source is the individual data instance. In medicine, however, shared context can occur at a higher level, the patient level. This idea is central to our contributions and will encourage the development of representations that are patient-specific. Such representations have the potential to be used in tasks that exploit patient similarity such as disease subgroup clustering and discovery. As a result of the process, medical practitioners may receive more interpretable outputs from networks.

In this work, we leverage electrocardiogram (ECG) signals to learn patient-specific representations in a self-supervised manner via contrastive learning. To do so, we exploit the fact that ECG signals summarize both temporal and spatial information. The latter can be understood in terms of projections of the same electrical signal onto multiple axes, also known as leads.

**Contributions.** Our contributions are the following:

1. We propose a family of patient-specific contrastive learning methods, entitled CLOCS, that exploit both temporal and spatial information present within ECG signals.

2. We show that CLOCS outperforms state-of-the-art methods, BYOL and SimCLR, when performing a linear evaluation of, and fine-tuning on, downstream tasks involving cardiac arrhythmia classification.

## 2 RELATED WORK

**Contrastive Learning.** In contrastive predictive coding, Oord et al. (2018) use representations of current segments to predict those of future segments. More recently, Tian et al. (2019) propose contrastive multi-view coding where multiple views of the same image are treated as 'shared context'. He et al. (2019); Chen et al. (2020); Grill et al. (2020) exploit the idea of instance discrimination (Wu et al., 2018) and interpret multiple views as stochastically augmented forms of the same instance. They explore the benefit of sequential data augmentations and show that cropping and colour distortions are the most important. These augmentations, however, do not trivially extend to the time-series domain. Shen et al. (2020) propose to create mixtures of images to smoothen the output distribution and thus prevent the model from being overly confident. Time Contrastive Learning (Hyvarinen & Morioka, 2016) performs contrastive learning over temporal segments in a signal and illustrate the relationship between their approach and ICA. In contrast to our work, they formulate their task as prediction of the segment index within a signal and perform limited experiments that do not exploit the noise contrastive estimation (NCE) loss. Bachman et al. (2019) Time Contrastive Networks (Sermanet et al., 2017) attempt to learn commonalities across views and differences across time. In contrast, our work focuses on identifying commonalities across *both* spatial and temporal components of data.

**Self-Supervision for Medical Time-Series.** Miotto et al. (2016) propose DeepPatient, a 3-layer stacked denoising autoencoder that attempts to learn a patient representation using electronic health record (EHR) data. Although performed on a large proprietary dataset, their approach is focused on EHRs and does not explore contrastive learning for physiological signals. Sarkar & Etemad (2020) apply existing self-supervised methods on ECG recordings in the context of affective computing. The methods implemented include defining pretext classification tasks such as temporal inversion, negation, time-warping, etc. Their work is limited to affective computing, does not explore contrastive learning, and does not exploit multi-lead data as we do. Lyu et al. (2018) explore a sequence to sequence model to learn representations from EHR data in the eICU dataset. In the process, they minimize the reconstruction error of the input time-series. Li et al. (2020) leverage the aforementioned unsupervised learning technique on a large clinical dataset, CPRD, to obtain uncertainty estimates for predictions.

## 3 BACKGROUND

### 3.1 CONTRASTIVE LEARNING

Assume the presence of a learner $f_\theta : x \in \mathbb{R}^D \to h \in \mathbb{R}^E$, parameterized by $\theta$, which maps a $D$-dimensional input, $x$, to an $E$-dimensional representation, $h$. Further assume the presence of an unlabelled dataset, $X \in \mathbb{R}^{N \times D}$, where $N$ is the total number of instances.

Each unlabelled instance, $x^i \in X$, is exposed to a set of transformations, $T_A$ and $T_B$, such that $x_A^i = T_A(x^i)$ and $x_B^i = T_B(x^i)$. Such transformations can consist of two different data augmentation procedures such as random cropping and flipping. These transformed instances now belong to an augmented dataset, $X' \in \mathbb{R}^{N \times D \times V}$, where $V$ is equal to the number of applied transformations. In contrastive learning, representations, $h_A^i = f_\theta(x_A^i)$ and $h_B^i = f_\theta(x_B^i)$, are said to share context. As a result of this shared context, these representations constitute a positive pair because (a) they are derived from the same original instance, $x^i$, and (b) the transformations applied to the original instance were class-preserving. Representations within a positive pair are encouraged to be similar to one another and dissimilar to representations of all other instances, $h_A^j, h_B^j \; \forall j \; j \neq i$. The similarity of these representations, $s(h_A^i, h_B^i)$, is quantified via a metric, $s$, such as cosine similarity. By encouraging high similarity between representations in the positive pair, the goal is to learn representations that are invariant to different transformations of the same instance.

## 4  METHODS

### 4.1  POSITIVE AND NEGATIVE PAIRS OF REPRESENTATIONS

Representations that are derived from the same *instance* are typically assumed to share context. This approach, however, fails to capture commonalities present across instances. In the medical domain, for example, multiple physiological recordings from the same patient may share context. It is important to note that if the multitude of physiological recordings associated with a patient were collected over large time-scales (e.g., on the order of years) and in drastically different scenarios (e.g., at rest vs. during a stress test), then the shared context across these recordings is likely to diminish. This could be due to changing patient demographics and disease profiles. With the previous caveat in mind, we propose to leverage commonalities present in multiple physiological recordings by redefining a positive pair to refer to representations of transformed instances that belong to the same *patient*. We outline how to arrive at these transformed instances next.

### 4.2  TRANSFORMATION OPERATORS

When choosing the transformation operators, $T$, that are applied to each instance, the principal desideratum is that they capture invariances in the ECG recording. Motivated by the observation that ECG recordings reflect both temporal and spatial information, we propose to exploit both temporal and spatial invariance. We provide an intuition for such invariances in Fig. 1.

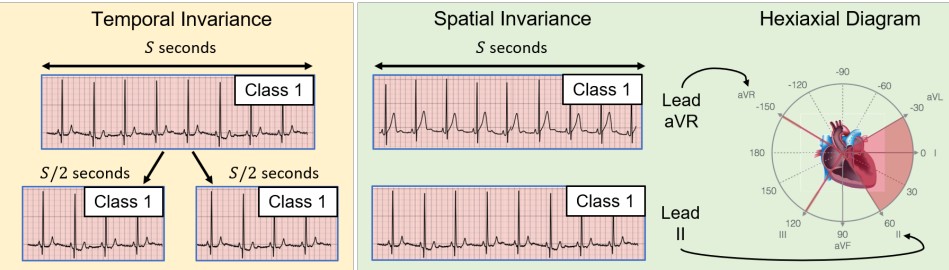

Figure 1: ECG recordings reflect both temporal and spatial information. This is because they measure the electrical activity of the heart using different leads (views) over time. **Temporal Invariance.** Abrupt changes to the ECG recording are unlikely to occur on the order of seconds, and therefore adjacent segments of shorter duration will continue to share context. **Spatial Invariance.** Recordings from different leads (at the same time) will reflect the same cardiac function, and thus share context.

As is pertains to temporal invariance (Fig. 1 left), we assume that upon splitting an ECG recording, associated with Class 1, into several segments, each of them remain associated with Class 1. We justify this assumption based on human physiology where abrupt changes in cardiac function (on the order of seconds) are unlikely to occur. If these segments were collected years apart, for example, our assumption may no longer hold. As for spatial invariance (Fig. 1 right), we leverage the hexiaxial diagram which illustrates the location of the leads relative to the heart. We assume that temporally-aligned ECG recordings from different leads (views) are associated with the same class. This is based on the idea that multiple leads (collected at the same time) will reflect the same underlying cardiac function. Occasionally, this assumption may not hold, if, for example, a cardiac condition afflicts a specific part of the heart, making it detectable by only a few leads. We now describe how to exploit these invariances for contrastive learning.

**Contrastive Multi-segment Coding (CMSC).** Given an ECG recording, $x^i$, with duration $S$ seconds, we can extract $V$ non-overlapping temporal segments, each with duration $S/V$ seconds. If $V = 2$, for example, $x^i_{t1} = T_{t1}(x^i)$ and $x^i_{t2} = T_{t2}(x^i)$ where $t$ indicates the timestamp of the temporal segment (see Fig. 1 left). We exploit temporal invariances in the ECG by defining representations of these adjacent and non-overlapping temporal segments as positive pairs.

**Contrastive Multi-lead Coding (CMLC).** Different projections of the same electrical signal emanating from the heart are characterized by different leads, $L$. For example, with two leads, $L1$ and $L2$, then $x^i_{L1} = T_{L1}(x^i)$ and $x^i_{L2} = T_{L2}(x^i)$ (see Fig. 1 right). We exploit spatial invariances in the ECG by defining temporally-aligned representations of these different projections as positive pairs.

**Contrastive Multi-segment Multi-lead Coding (CMSMLC).** We simultaneously exploit both temporal and spatial invariances in the ECG by defining representations of non-overlapping temporal segments and different projections as positive pairs. For example, in the presence of two temporal segments with timestamps, $t1$ and $t2$, that belong to two leads, $L1$ and $L2$, then $x_{t1,L1}^i = T_{t1,L1}(x^i)$ and $x_{t2,L2}^i = T_{t2,L2}(x^i)$.

### 4.3 PATIENT-SPECIFIC NOISE CONTRASTIVE ESTIMATION LOSS

Given our patient-centric definition of positive pairs, we propose to optimize a patient-specific noise contrastive estimation loss. More formally, Given a mini-batch of $K$ instances, we apply a pair of transformation operators and generate $2K$ transformed instances (a subset of which is shown in Fig. 2). We encourage a pair of representations, $h_A^i$ and $h_B^k$, $i, k \in P$, from the same patient, $P$, to be similar to one another and dissimilar to representations from other patients. We quantify this similarity using the cosine similarity, $s$, with a temperature scaling parameter, $\tau$, (see Eq. 4) as is performed in (Tian et al., 2019; Chen et al., 2020). We extend this to all representations in the mini-batch to form a similarity matrix of dimension $K \times K$. In this matrix, we identify positive pairs by associating each instance with its patient ID. By design, this includes the diagonal elements and results in the loss shown in Eq. 2. If the same patient reappears within the mini-batch, then we also consider off-diagonal elements, resulting in the loss shown in Eq. 3. The frequency of these off-diagonals is inconsistent due to the random shuffling of data. We optimize the objective function in Eq. 1 for all pairwise combinations of transformation operators, $T_A$ and $T_B$, where we include Eq. 2 and Eq. 3 twice to consider negative pairs in both views.

$$\mathcal{L} = \mathbb{E}_{T_A, T_B} \left[ \mathcal{L}_{diag}^{h_A, h_B} + \mathcal{L}_{diag}^{h_B, h_A} + \mathcal{L}_{off-diag}^{h_A, h_B} + \mathcal{L}_{off-diag}^{h_B, h_A} \right] \quad (1)$$

$$\mathcal{L}_{diag}^{h_A, h_B} = -\mathbb{E}_{i \in P} \left[ \log \frac{e^{s(h_A^i, h_B^i)}}{\sum_j e^{s(h_A^i, h_B^j)}} \right] \quad (2)$$

$$\mathcal{L}_{off-diag}^{h_A, h_B} = -\mathbb{E}_{i,k \in P} \left[ \log \frac{e^{s(h_A^i, h_B^k)}}{\sum_j e^{s(h_A^i, h_B^j)}} \right] \quad (3) \quad s(h_A^i, h_B^i) = \frac{f_\theta(x_A^i) \cdot f_\theta(x_B^i)}{\|f_\theta(x_A^i)\| \|f_\theta(x_B^i)\|} \frac{1}{\tau} \quad (4)$$

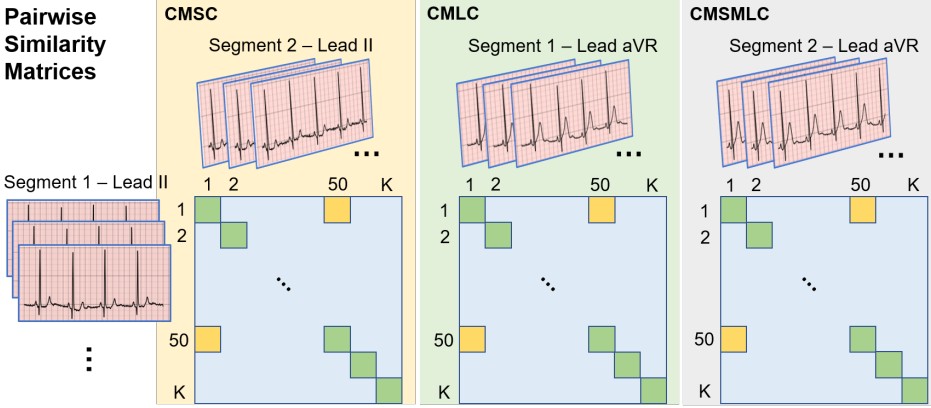

Figure 2: Similarity matrix for a mini-batch of $K$ instances in (Left) **Contrastive Multi-segment Coding**, (Centre) **Contrastive Multi-lead Coding**, and (Right) **Contrastive Multi-segment Multi-lead Coding**. Additional matrices would be generated based on all pairs of applied transformation operators, $T_A$ and $T_B$. Exemplar transformed ECG instances are illustrated along the edges. To identify positive pairs, we associate each instance with its patient ID. By design, diagonal elements (green) correspond to the same patient, contributing to Eq. 2. Similarly, instances 1 and 50 (yellow) belong to the same patient, contributing to Eq. 3. The blue area corresponds to negative examples as they pertain to instances from different patients.

## 5 EXPERIMENTAL DESIGN

### 5.1 DATASETS

We conduct our experiments using PyTorch (Paszke et al., 2019) on four ECG datasets that include cardiac arrhythmia labels. **PhysioNet 2020** (Perez Alday et al., 2020) consists of 12-lead ECG recordings from 6,877 patients alongside 9 different classes of cardiac arrhythmia. Each recording can be associated with multiple labels. **Chapman** (Zheng et al., 2020) consists of 12-lead ECG recordings from 10,646 patients alongside 11 different classes of cardiac arrhythmia. As is suggested by Zheng et al. (2020), we group these labels into 4 major classes. **PhysioNet 2017** (Clifford et al., 2017) consists of 8,528 single-lead ECG recordings alongside 4 different classes. **Cardiology** (Hannun et al., 2019) consists of single-lead ECG recordings from 328 patients alongside 12 different classes of cardiac arrhythmia. An in-depth description of these datasets can be found in Appendix A.1.

All datasets were split into training, validation, and test sets according to patient ID using a 60, 20, 20 configuration. In other words, patients appeared in only one of the sets. The exact number of instances used during self-supervised pre-training and supervised training can be found in Appendix A.2.

### 5.2 PRE-TRAINING IMPLEMENTATION

We conduct our pre-training experiments on the training set of two of the four datasets: PhysioNet 2020 and Chapman. We chose these datasets as they contain multi-lead data. In **CMSC**, we extract a pair of non-overlapping temporal segments of $S = 2500$ samples. This is equivalent to either 10 or 5 seconds worth of ECG data from the Chapman and PhysioNet 2020 datasets, respectively. Therefore, our model is presented with a mini-batch of dimension $K \times S \times 2$ where $K$ is the batchsize, and $S$ is the number of samples. In **CMLC**, we explore two scenarios with a different number of leads corresponding to the same instance. Our mini-batch dimension is $K \times S \times L$, where $L$ is the number of leads. Lastly, in **CMSMLC**, we incorporate an additional temporal segment in each mini-batch. Therefore, our mini-batch dimension is $K \times 2S \times L$. To ensure a fair comparison between all methods, we expose them to an equal number of patients and instances during training. In CMLC or CMSMLC, we either pre-train using 4 leads (II, V2, aVL, aVR) or all 12 leads. We chose these 4 leads as they cover a large range of axes.

### 5.3 EVALUATION ON DOWNSTREAM TASK

We evaluate our pre-trained methods in two scenarios. In **Linear Evaluation of Representations**, we are interested in evaluating the utility of the fixed feature extractor in learning representations. Therefore, the pre-trained parameters are frozen and multinomial logistic regression is performed on the downstream supervised task. In **Transfer Capabilities of Representations**, we are interested in evaluating the inductive bias introduced by pre-training. Therefore, the pre-trained parameters are used as an initialization for training on the downstream supervised task.

### 5.4 BASELINES

We compare our pre-training methods to networks that are initialized randomly (**Random Init.**), via supervised pre-training (**Supervised**), or via a multi-task pre-training mechanism introduced specifically for ECG signals (**MT-SSL**) (Sarkar & Etemad, 2020). We also compare to **BYOL** (Grill et al., 2020) and **SimCLR** (Chen et al., 2020), which encourage representations of instances and their perturbed counterparts to be similar to one another, with the aim of learning transformation-invariant representations that transfer well. As SimCLR has been shown to be highly dependent on the choice of perturbations, we explore the following time-series perturbations (see Appendix B for visualizations):

- **Gaussian** - we add $\epsilon \sim \mathcal{N}(0, \sigma)$ to the time-series signal where we chose $\sigma$ based on the amplitude of the signal. This was motivated by the work of Han et al. (2020) who recently showed the effect of additive noise on ECG signals.
- **Flip** - we flip the time-series signal temporally (**Flip**$_Y$), reversing the arrow of time, or we invert the time-series signal along the x-axis (**Flip**$_X$).
- **SpecAugment** (Park et al., 2019) - we take the short-time Fourier transform of the time-series signal, generating a spectrogram. We then mask either temporal (**SA**$_t$) or spectral (**SA**$_f$) bins

of varying widths before converting the spectrogram to the time domain. We also explore the application of sequential perturbations to the time-series signal.

## 5.5 HYPERPARAMETERS

During self-supervised pre-training, we chose the temperature parameter, $\tau = 0.1$, as per Chen et al. (2020). For BYOL, we chose the decay rate, $\tau_d = 0.90$, after experimenting with various alternatives (see Appendix F). We use the same network architecture for all experiments. Further implementation details can be found in Appendix C.

## 6 EXPERIMENTAL RESULTS

### 6.1 LINEAR EVALUATION OF REPRESENTATIONS

In this section, we evaluate the utility of the self-supervised representations learned using four leads on a downstream linear classification task. In Table 1, we show the test AUC on Chapman and PhysioNet 2020 using 50% of the labelled data ($F = 0.5$) after having learned representations, with dimension $E = 128$, using the same two datasets.

We show that CMSC outperforms BYOL and SimCLR on both datasets. On the Chapman dataset, CMSC and SimCLR achieve an AUC = 0.896 and 0.738, respectively, illustrating a 15.8% improvement. Such a finding implies that the representations learned by CMSC are richer and thus allow for improved generalization. We hypothesize that this is due to the setup of CMSC whereby the shared context is across segments (temporally) and patients. Moreover, we show that CLOCS (all 3 proposed methods) outperforms SimCLR in 100% of all conducted experiments, even when pre-training and evaluating with all 12 leads (see Appendix D).

Table 1: Test AUC of the linear evaluation of the representations at $F = 0.5$, after having pre-trained on Chapman or PhysioNet 2020 with $E = 128$. Pre-training and evaluating multi-lead datasets* using 4 leads (II, V2, aVL, aVR). Mean and standard deviation are shown across 5 seeds.

| Dataset | Chapman* | PhysioNet 2020* |
|---|---|---|
| MT-SSL | $0.677 \pm 0.024$ | $0.665 \pm 0.015$ |
| BYOL | $0.643 \pm 0.043$ | $0.595 \pm 0.018$ |
| SimCLR | $0.738 \pm 0.034$ | $0.615 \pm 0.014$ |
| CMSC | $\mathbf{0.896 \pm 0.005}$ | $\mathbf{0.715 \pm 0.033}$ |
| CMLC | $0.870 \pm 0.022$ | $0.596 \pm 0.008$ |
| CMSMLC | $0.847 \pm 0.024$ | $0.680 \pm 0.008$ |

### 6.2 EFFECT OF PERTURBATIONS ON PERFORMANCE

So far, we have presented CLOCS without having incorporated any perturbations during pre-training. However, contrastive learning methods, and in particular SimCLR, are notorious for their over-dependence on the choice of perturbations. To explore this dependence, we apply a diverse set of stochastic perturbations, $P$, (see Appendix B) during pre-training and observe its effect on generalization performance. We follow the setup introduced by Chen et al. (2020) and apply either a **single perturbation** to each instance, $x^i$, whereby $x_1^i = P_1(x^i)$, or **sequential perturbations** whereby $x_{1,2}^i = P_2(P_1(x^i))$.

We apply such perturbations while pre-training with SimCLR or CMSC on PhysioNet 2020 using 4 leads and, in Fig. 3, illustrate the test AUC in the linear evaluation scenario. We show that, regardless of the type and number of perturbations, CMSC continues to outperform SimCLR. For example, the *worst-performing* CMSC implementation ($\text{Flip}_Y$) results in an AUC = 0.661 which is still greater than the *best-performing* SimCLR implementation (Gaussian $\rightarrow \text{SA}_t$) with an AUC = 0.636. In fact, we find that pre-training with CMSC *without* applying any perturbations (see Table 1) still outperforms the best-performing SimCLR implementation. Such a finding suggests that CMSC's already strong performance is more likely to stem from its redefinition of the 'shared context' to include both time and patients than from the choice of perturbations.

### 6.3 TRANSFER CAPABILITIES OF REPRESENTATIONS

In this section, we evaluate the utility of initializing a network for a downstream task with parameters learned via self-supervision using four leads. In Table 2, we show the test AUC on downstream datasets at $F = 0.5$ for the various self-supervised methods with $E = 128$.

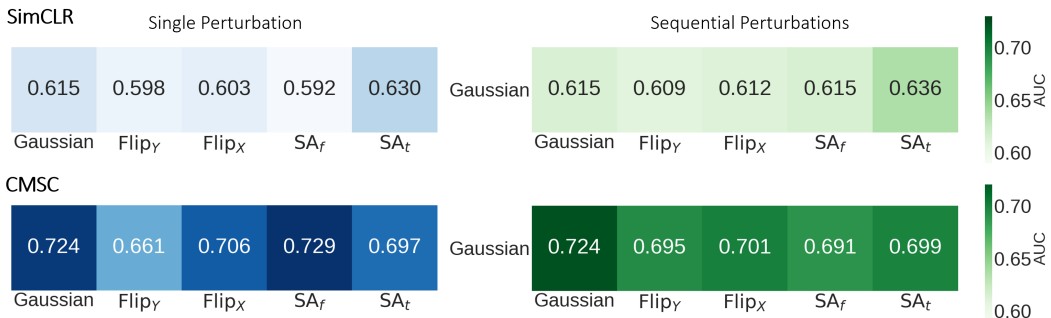

Figure 3: Effect of single (blue) and sequential (green) perturbations applied to the (top) SimCLR and (bottom) CMSC implementations on linear evaluation. Sequential perturbations involve a Gaussian perturbation followed by one of the remaining four types. Pre-training and evaluation was performed on PhysioNet 2020 using 4 leads. Evaluation was performed at $F = 0.5$ and results are averaged across 5 seeds. We show that CMSC outperforms SimCLR regardless of the applied perturbation.

We show that, with a few exceptions, self-supervision is advantageous relative to a Random Initialization. This can be seen by the higher AUC achieved by the former relative to the latter. We also show that, depending on the downstream dataset, either CMSC or CMSMLC outperform BYOL and SimCLR. For example, when pre-training on Chapman and fine-tuning on Cardiology, CMSMLC achieves an $AUC = 0.717$, a $4.1\%$ improvement compared to SimCLR. This implies that by encouraging representations across space, time, and patients to be similar to one another, networks are nudged into a favourable parameter space. In Appendix E.1, we extend these findings and illustrate that CLOCS outperforms SimCLR in at least $75\%$ of all experiments conducted, on average. When pre-training, fine-tuning, and evaluating using all 12 leads, we show that CMSC outperforms all other methods in at least $90\%$ of all experiments conducted (see Appendix E.2).

Table 2: Test AUC in the fine-tuning scenario at $F = 0.5$, after having pre-trained on Chapman or PhysioNet 2020 with $E = 128$. Pre-training, fine-tuning, and evaluating multi-lead datasets* using 4 leads. Mean and standard deviation are shown across 5 seeds.

| Pretraining Dataset | Chapman* | | | PhysioNet 2020* | | |
|---|---|---|---|---|---|---|
| Downstream Dataset | Cardiology | PhysioNet 2017 | PhysioNet 2020* | Cardiology | PhysioNet 2017 | Chapman* |
| Random Init. | $0.678 \pm 0.011$ | $0.763 \pm 0.005$ | $0.803 \pm 0.008$ | $0.678 \pm 0.011$ | $0.763 \pm 0.005$ | $0.907 \pm 0.006$ |
| Supervised | $0.684 \pm 0.015$ | $0.799 \pm 0.008$ | $0.827 \pm 0.001$ | $0.730 \pm 0.002$ | $0.810 \pm 0.009$ | $0.954 \pm 0.003$ |
| *Self-supervised Pre-training* | | | | | | |
| MT-SSL | $0.650 \pm 0.009$ | $0.741 \pm 0.012$ | $0.774 \pm 0.010$ | $0.661 \pm 0.011$ | $0.746 \pm 0.016$ | $0.923 \pm 0.007$ |
| BYOL | $0.678 \pm 0.021$ | $0.748 \pm 0.014$ | $0.802 \pm 0.013$ | $0.674 \pm 0.022$ | $0.757 \pm 0.010$ | $0.916 \pm 0.009$ |
| SimCLR | $0.676 \pm 0.011$ | $0.772 \pm 0.010$ | $0.823 \pm 0.011$ | $0.658 \pm 0.027$ | $0.762 \pm 0.009$ | $0.923 \pm 0.010$ |
| CMSC | $0.695 \pm 0.024$ | $0.773 \pm 0.013$ | $\mathbf{0.830 \pm 0.002}$ | $\mathbf{0.714 \pm 0.014}$ | $0.760 \pm 0.013$ | $\mathbf{0.932 \pm 0.008}$ |
| CMLC | $0.665 \pm 0.016$ | $0.767 \pm 0.013$ | $0.810 \pm 0.011$ | $0.675 \pm 0.013$ | $0.762 \pm 0.007$ | $0.910 \pm 0.012$ |
| CMSMLC | $\mathbf{0.717 \pm 0.006}$ | $\mathbf{0.774 \pm 0.004}$ | $0.814 \pm 0.009$ | $0.698 \pm 0.011$ | $\mathbf{0.774 \pm 0.012}$ | $0.930 \pm 0.012$ |

## 6.4 DOING MORE WITH LESS LABELLED DATA

Having established that self-supervision can nudge networks to a favourable parameter space, we set out to investigate whether such a space can lead to strong generalization with less labelled data in the downstream task. In Fig. 4, we illustrate the validation AUC of networks initialized randomly or via CMSC and fine-tuned on two different datasets.

We find that fine-tuning a network based on a CMSC initialization drastically improves data-efficiency. In Fig. 4a, we show that a network initialized with CMSC and exposed to only $25\%$ of the labelled data outperforms one that is initialized randomly and exposed to $100\%$ of the labelled data. This can be seen by the consistently higher AUC during, and at the end of, training. A similar outcome can be seen in Fig. 4b. This suggests that self-supervised pre-training exploits data efficiently such that it can do more with less on downstream classification tasks.

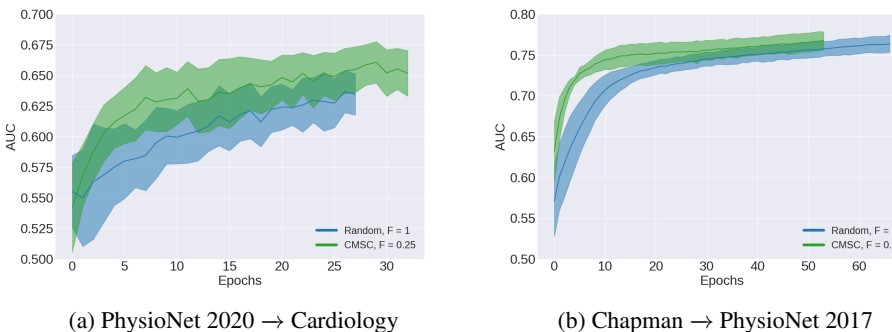

(a) PhysioNet 2020 → Cardiology (b) Chapman → PhysioNet 2017

Figure 4: Validation AUC of a network initialized randomly or via CMSC and which is exposed to different amounts of labelled training data, $F$. Results are averaged across 5 seeds. Shaded area represents one standard deviation.

## 6.5 EFFECT OF EMBEDDING DIMENSION, $E$, AND AVAILABILITY OF LABELLED DATA, $F$

The dimension of the representation learned during self-supervision and the availability of labelled training data can both have an effect on model performance. In this section, we investigate these claims. In Figs. 5a and 5b, we illustrate the test AUC for all pre-training methods as a function of $E = (32, 64, 128, 256)$ and $F = (0.25, 0.50, 0.75, 1)$.

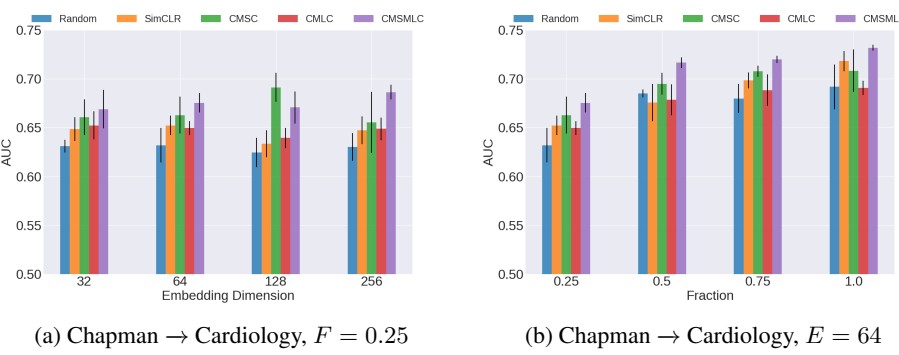

(a) Chapman → Cardiology, $F = 0.25$ (b) Chapman → Cardiology, $E = 64$

Figure 5: Effect of (a) embedding dimension, $E$, and (b) labelled fraction, $F$, on the test AUC when pre-training on Chapman and fine-tuning on Cardiology. Results are averaged across 5 seeds. Error bars represent one standard deviation.

In Fig. 5a, we show that networks initialized randomly or via SimCLR are not significantly affected by the embedding dimension. This can be seen by the AUC $\approx 0.63$ and $\approx 0.65$, for these two methods across all values of $E$. In contrast, the embedding dimension has a greater effect on CMSC where AUC $\approx 0.66 \rightarrow 0.69$ as $E = 32 \rightarrow 128$. This implies that CMSC is still capable of achieving strong generalization performance despite the presence of few labelled data ($F = 0.25$). We hypothesize that the strong performance of CMSC, particularly at $E = 128$, is driven by its learning of patient-specific representations (see Appendix G) that cluster tightly around one another, a positive characteristic especially when such representations map to the same downstream class.

In Fig. 5b, we show that increasing the amount of labelled training data benefits the generalization performance of all methods. This can be seen by the increasing AUC values as $F = 0.25 \rightarrow 1$. We also show that at all fraction values, CMSMLC outperforms its counterparts. For example, at $F = 1$, CMSMLC achieves an AUC $= 0.732$ whereas SimCLR achieves an AUC $= 0.718$. Such superiority still holds at $F = 0.25$ where the two methods achieve an AUC $= 0.675$ and $0.652$, respectively. This outcome emphasizes the robustness of CMSMLC to scarce labelled training data.

## 6.6    CLOCS LEARNS PATIENT-SPECIFIC REPRESENTATIONS

We redefined 'shared context' to refer to representations from the same patient, which in turn should produce patient-specific representations. To validate this hypothesis, we calculate the pairwise Euclidean distance between representations of the same patient (Intra-Patient) and those of different patients (Inter-Patient). On average, the former should be smaller than the latter. In Fig. 6, we illustrate the two distributions associated with the intra and inter-patient distances at $E = 128$. We also find that increasing the embedding dimension shifts these distributions to higher values (see Fig 9).

We show that these two distributions have large mean values and overlap significantly when implementing SimCLR, as seen in Fig. 6a. This is expected as SimCLR is blind to the notion of a patient. In contrast, when implementing CMSC, the intra-patient distances are lower than those found in SimCLR, as seen in Fig. 6b. Moreover, the intra and inter-patient distributions are more separable. This implies that pre-training with CMSC leads to patient-specific representations. We note that this phenomenon takes place while concomitantly learning better representations, as observed in previous sections.

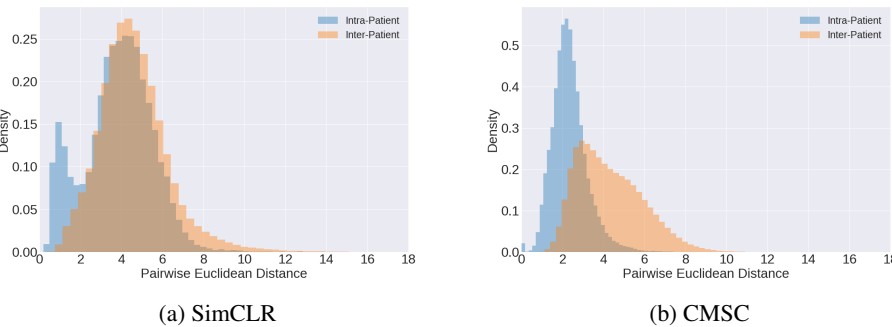

(a) SimCLR                                                    (b) CMSC

Figure 6: Distribution of pairwise Euclidean distance between representations ($E = 128$) belonging to the same patient (Intra-Patient) and those belonging to different patients (Inter-Patient). Self-supervision was performed on PhysioNet 2020. Notice the lower average intra-patient distance and improved separability between the two distributions with CMSC than with SimCLR.

## 7    DISCUSSION AND FUTURE WORK

In this paper, we proposed a family of self-supervised pre-training mechanisms, entitled CLOCS, based on contrastive learning for physiological signals. In the process, we encourage representations across segments (temporally) and leads (spatially) that correspond to instances from the same patient to be similar to one another. We show that our methods outperform the state-of-the-art methods, BYOL and SimCLR, when performing a linear evaluation of, and fine-tuning on, downstream tasks. This conclusion also holds when applying a range of perturbations and when pre-training and evaluating with a different number of leads. We now elucidate several avenues worth exploring.

**Quantifying patient similarity.** We have managed to learn patient-specific representations. These representations can be used to quantify patient-similarity in order to assist with diagnosis or gain a better understanding of a diseased condition. Validation of these representations can be performed by comparing known similar patients.

**Multi-modal transfer.** We transferred parameters from one task to another that shared the same input modality, the ECG. Such data may not always be available for self-supervision. An interesting path would be to explore whether contrastive self-supervision on one modality can transfer well to another modality.

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
