# OpenReview forum: "CLOCS: Contrastive Learning of Cardiac Signals Across Space, Time, and Patients"
_ICLR.cc/2021/Conference — Reject_

### Official Review · AnonReviewer4 · 2020-10-23
**Questions about intuitions of Contrastive Multi-lead Coding**

**Rating:** 4
**Confidence:** 5

**Review:**

This paper proposes to use contrastive learning to learn representations from cardiac signals (ECGs). The model incorporates ECG domain knowledge, patient-specific, and relationships between multiple leads (channels), in the learning process. The targeted task is very important, enormous ECGs are collected and stored, but seldom people are mining them as they are unlabelled. This paper might reinvigorate them.

Major:

However, some of the domain knowledge-based intuitions are unclear (might even be wrong). For example, for Contrastive Multi-lead Coding, we know that ECGs are 2D projections from the heart's 3D electricity activities. The similarity between different leads is related to the projection surface. What if two leads are orthogonally projected? They should be unrelated at all. Thus would even bring the noise in the model. Maybe this is the reason why CMSC in Table 1 is the best, but adding Multi-lead Coding (CMSMLC) drops the performance.

In fact, unsupervised learning usually needs (much) larger datasets. But all four experimental datasets are actually well annotated (relatively small) ECG databases. It seems more reasonable to experiment on a larger unlabelled dataset, such as the MIMIC-III waveform database (https://physionet.org/content/mimic3wdb/1.0/). And it might give more convincing and desired results.

Others:

(1) This paper mainly focuses on design positive and negative pairs from the perspective of ECG domain knowledge - between leads, within subject. The technical contribution looks limited.

(2) Some related papers about contrastive learning on physiological signals.

Published (you might reconsider the last sentence in Related Work.):

[1] Banville, Hubert, et al. "Self-supervised representation learning from electroencephalography signals." 2019 IEEE 29th International Workshop on Machine Learning for Signal Processing (MLSP). IEEE, 2019.

Preprint:

[2] Cheng, Joseph Y., et al. "Subject-aware contrastive learning for biosignals." arXiv preprint arXiv:2007.04871 (2020).

[3] Banville, Hubert, et al. "Uncovering the structure of clinical EEG signals with self-supervised learning." arXiv preprint arXiv:2007.16104 (2020).

(3) Multi-lead ECG is more common in clinical usage. Can this method be extended to learn representations from multi-lead ECG?

---

> ### Author Response · Authors · 2020-11-18
> **Response to Reviewer 4 - Round 1**
>
> We thank the reviewer for taking the time and effort to review the manuscript and for providing us with valuable feedback. We address your comments below.
>
> **KNOWLEDGE-BASED INTUITIONS**
> We would like to respectfully disagree that our knowledge-based intuitions are wrong. To clarify any potential misunderstandings, we have modified Sec. 4 to provide further intuition behind our methods (CMSC and CMLC in particular). We urge the reviewer to read this section once more and please let us know if anything is unclear.
>
> In the introduction we provided readers with a high-level description of the concept of ‘leads’ as projections of the same electrical signal onto multiple axes. We opted for this accurate, albeit simplistic definition, of the concept of leads to avoid medical jargon which could confuse readers. As you mentioned, the similarity of leads is related to the axis that each lead occupies. A visual guide of such axes, also known as the hexiaxial system, can be found at this link (https://www.ncbi.nlm.nih.gov/pmc/articles/PMC1122339/pdf/415.pdf).
>
> We would like to clarify that our motivation for the Contrastive Multi-lead Coding (CMLC) approach is based on the notion that multiple ECG leads (corresponding to the same temporal segment) are likely to reflect the same cardiac arrhythmia label. Take the following example; without loss of generality, let us assume we have a 12-lead ECG segment with a duration of 10 seconds that reflects a normally functioning heart. This implies that the entire 12-lead ECG segment would be associated with the label ‘Normal Sinus Rhythm’ (NSR) to indicate normalcy. Now, if we were to take two of these 12 leads, for example, such as Lead I and Lead II and observe the shape (morphology) of their ECG signals, both of these signals would exhibit a normal shape. In other words, ECG signals across space (leads) map to the same label (NSR). This explanation, in turn, supports one of our ongoing themes throughout the paper, that of class-preserving transformations.
>
> Despite the aforementioned, we do concede to the idea that occasionally the same cardiac condition may not be reflected in all leads. In such cases, attracting representations of different leads to one another might be detrimental. We do believe this inappropriate attraction explains some of the relatively poorer performance exhibited by CMLC and CMSMLC relative to CMSC on certain datasets. Our initial concern with this inappropriate attraction across leads had motivated our pre-training experiments with fewer than all 12 leads (e.g., only 4). These extensive results can be found in Appendices D and E. However, we found that on some datasets, 4-lead pre-training outperformed its 12-lead counterpart and on other datasets, the opposite was true.
>
> **Pre-training on MIMICIII WFDB**
> Although self-supervised learning has been shown to benefit from large datasets (in computer vision), the definition of 'large' when dealing with ECG time-series data is just different. Until this year (2020), there were no large publicly-available 12-lead ECG datasets. 'Large' in our domain is limited to 10,000 patients, especially when working with public data. Please note that even the MIMICIII WFDB does not contain such data, which would be required for our pre-training methods, in particular CMLC and CMSMLC, which necessitate the availability of multi-lead data. Having said that, we have illustrated the superiority of our methods despite this 'limited' data regime that you describe. Larger datasets are only expected to improve this further.
>
> We hope the above responses and the modified version of the manuscript have addressed your concerns.

---

### Official Review · AnonReviewer1 · 2020-10-29

**Rating:** 4
**Confidence:** 5

**Review:**

Summary:
This work presents a new self-supervised training framework for multi-channel ECG signals. The authors use contrastive learning by exploiting the fact that a single patient can generate multiple ECG signals, and there are multiple views (i.e. leads) for the same ECG signals. Compared to popular self-supervised training methods BYOL and SimCLR, the proposed method shows superior performance on the arrhythmia classification task for 4 different datasets in various scenarios.

Pros:
- The proposed approach is easy to follow, makes much sense, and is well-motivated based on the domain-specific knowledge of ECG signals.
- The proposed approach outperforms popular baselines (although borrowed from the vision community) for the arrhythmia classification task on various combinations of experiment setups.

Cons:
- The most critical shortcoming of this paper is that the baselines, although widely known, are borrowed from the computer vision community only, which is the reason for rating 4. There are already several papers proposing self-supervised learning methods for time-series data such as audio signals, ECGs and EEGs [1, 2, 3, 4, 5]. The authors should compose a stronger baseline set to demonstrate the value of this work. (Audio signals are usually single-channel unlike ECG signals, but CMSC alone already demonstrates good performance, hence methods such as wave2vec could be a strong baseline)
- The third paragraph of section 6.5, where the authors claim the superiority of CMSC across various embedding dimensions seems like a stretch. The AUC difference of SimCLR (which is 0.02) and that of CMSC (0.03) is not that large, and CMSC even underperforms when the embedding dimension is 256. The proposed method has already shown good performance in various cases, so there is probably no need to make a stretched claim.

Reference
1. Schneider, S., Baevski, A., Collobert, R. and Auli, M., 2019. wav2vec: Unsupervised pre-training for speech recognition. arXiv preprint arXiv:1904.05862.
2. Baevski, A., Zhou, H., Mohamed, A. and Auli, M., 2020. wav2vec 2.0: A framework for self-supervised learning of speech representations. arXiv preprint arXiv:2006.11477.
3. Banville, H., Moffat, G., Albuquerque, I., Engemann, D.A., Hyvärinen, A. and Gramfort, A., 2019, October. Self-supervised representation learning from electroencephalography signals. In 2019 IEEE 29th International Workshop on Machine Learning for Signal Processing (MLSP) (pp. 1-6). IEEE.
4. Sarkar, P. and Etemad, A., 2020. Self-supervised ecg representation learning for emotion recognition. arXiv preprint arXiv:2002.03898.
5. Cheng, J.Y., Goh, H., Dogrusoz, K., Tuzel, O. and Azemi, E., 2020. Subject-aware contrastive learning for biosignals. arXiv preprint arXiv:2007.04871.

---

> ### Author Response · Authors · 2020-11-18
> **Response to Reviewer 1 - Round 1**
>
> We thank the reviewer for taking the time and effort to review the manuscript and for providing us with valuable feedback. We address your comments below.
>
> **BASELINES**
> We understand that BYOL and SimCLR had initially been introduced in the context of computer vision but their application is much broader than that. Even in our experiments, we show that SimCLR can be, in fact, quite competitive with the family of methods encapsulated by CLOCS. As for your suggestions regarding other baseline methods, we implemented Wav2Vec2.0 (based on this implementation https://github.com/pytorch/fairseq/tree/master/examples/wav2vec) on our ECG time-series data as a pre-training method and could not manage to get the network to converge. As a result, we did not pursue this avenue further.
>
> **CLAIMS IN SECTION 6.5**
> To clarify, the claim we make in section 6.5 is less about the superiority of CMSC and more about the effect of the embedding dimension, E, on the various methods. We are simply trying to show that E has a greater effect on CMSC, where performance eventually peaks at E=128.
>
> We urge the reviewer to read over the modified version of the manuscript. We hope the above responses and the modified version of the manuscript have addressed your concerns.

---

### Official Review · AnonReviewer3 · 2020-10-29
**Simple but elegant**

**Rating:** 7
**Confidence:** 4

**Review:**

The paper describes a method for unsupervised learning of patient representations from ECG data.
Using contrastive learning, ECG recordings from different time periods and different leads are optimized to be similar for the same patient and different for all the other patients.
The method is evaluated on several datasets, showing improvement over random initialization and an alternative method using data permutations.

The method is simple and a relatively minor extension of contrastive learning to time series.
But it is elegant, shows good results over alternative approaches and can be a useful solution for time series.

The paper doesn't currently  contain a description of the actual input and the network architecture. There are 29 pages of appendices and these have some additional details, but the main paper should contain some of this important information as well.

Equations 1-4 need to come with some explanation. At the moment, several variables there are left undefined and the overall intuition behind structuring the loss equations should be explained.

Could these additional objectives or data augmentations be applied during the fine-tuning phase as well? If so, would that reduce the benefit of unsupervised pretraining?

---

> ### Author Response · Authors · 2020-11-18
> **Response to Reviewer 3 - Round 1**
>
> We thank the reviewer for taking the time and effort to review the manuscript and for providing us with valuable feedback. We address your comments below.
>
> **EXPLANATION OF EQUATIONS 1-4**
> We rewrite the entirety of Section 4 in order to improve clarity. More specifically, we introduce Fig. 1 to better explain the intuition behind our proposed methods (CMSC and CMLC). We then leverage Fig. 2 to better explain equations 1-4 and how they complement one another. Once you have read this section, please let us know if the equations (or anything else) remain unclear.
>
> **DATA AUGMENTATION ON DOWNSTREAM TASK**
> As for the data augmentations that we experiment with (e.g., Gaussian noise, Flip, and SpecAugment), they can also be applied during the fine-tuning phase. To better determine the effect of such downstream data augmentations on the benefit of self-supervised pre-training, one could conduct, and compare the results of, the following experiments: 1) Random Init. + Downstream Data Augmentations and 2) CLOCS + Downstream Data Augmentations. However, training medical time-series signals with data augmentations is non-trivial since the data augmentations usually have to be domain-specific and requires a significant amount of hyper-parameter tuning. For example, we experimented with Gaussian noise and SpecAugment as a data augmentation technique with a random initialization and could not manage to obtain a network that converges. Identifying appropriate data augmentation for downstream fine-tuning is an interesting problem but beyond the scope of our paper.
>
> We hope the above responses and the modified version of the manuscript have addressed your concerns.

---

### Official Review · AnonReviewer2 · 2020-11-06
**interesting topic; lack of insight; unsupported claims**

**Rating:** 5
**Confidence:** 3

**Review:**

This paper proposes a contrastive learning method for cardiac signals.

This strong points of the paper are the following:
1. It presents its problem clearly. The problem it targets is an impactful problem in the medical domain.
2. The method it proposes is straight forward and easily understandable.
3. It demonstrates the advantage of the proposed method on a real dataset by comparing against the SOTA CL methods like BYOL and SimCLR.

The paper also has weaknesses.
1. unsupported claims In the abstract.
 it is claimed that "our training procedure naturally generates patient-specific representations that can be used to quantify patient-similarity". Then the next time where the "patient-similarity" appears is in the discussion. It says "We have managed to learn patient-specific representations....". However, in the main paper, I can not find any evidence showing the representation it learns is capturing the patient-similarity. It is unacceptable to claim something by just claiming it.
2.unclear contributions.
 In the introduction, the author is listing their methods as contributions. Usually, the contribution should be something achieved by you while not previously achieved. It could the new best performance, a new solution to a problem, a new angle of viewing the problem. The method itself alone hardly can be the contributions. I can propose any algorithms. They can not be contributions if they are not useful. So I suggest the author think about elaborating on what are your contributions.
3.not sufficient description of the task.
 In the whole paper, the only thing I get about the task is that the author focuses on is a classification task related to cardiac arrhythmia. This is far from sufficient. The author should provide the backgrounds of the task. What are the extra 11 or 4 classes of cardiac arrhythmia? Is the task label also time series? or it is a global label that does change across time? For most of the readers from the ML community, they do not have a background in cardiac signals. To let the audience understand the tasks, I suggest the author to including visualization about the class label and the cardiac signals. Currently, I have no idea about what the task exactly is.
4.lack of insight.
 Currently, the paper is mainly written in a way like I did A,B,C,D; they are better than previous method E,F; see my numbers. Here are my questions. Why doing A,B,C,D? For example, why Constrasitve Multi-segments Coding make sense? This question is also related to my previous point. What is your class label? If it is also a time series, considering the following case. At time step t1, the patient is normal, while at time step t2, the patient is in some state of cardiac arrhythmia. Why it makes sense to encourage the representations at t1 and t2 to be similar?
Actually, the experimental result is interesting. From Table 2, I notice that the best performance is achieved by either CMSC or CMSMLC. It means that contrastive multi-segments across time is useful. It contradicts my intuition in the last paragraph. Can the author give any explanation?
There are more questions about the results the author showed. For example, in Figure 4(a), CMSC has an abnormally high AUC when embedding dimension = 128. Why does it happen?

I hope the author can include more insight or deep analysis in the paper to help the reader learn more from this paper.
The paper has a good topic, good methods, and good experiments. However, it still needs some effort to be a good paper. I hope my comments can help the author improve the paper and make it good work. My rating is not final. I will raise the score if my main concerns are addressed.

---

> ### Author Response · Authors · 2020-11-18
> **Response to Reviewer 2 - Round 1**
>
> We thank the reviewer for taking the time and effort to review the manuscript and for providing us with valuable feedback. We address your comments below.
>
> **CLAIMS ABOUT PATIENT-SPECIFIC REPRESENTATIONS**
> In our original submission, we had relegated the section that contained figures to support our claim that CLOCS learns ‘patient-specific representations’ to the Appendix (Appendix G). Given such relegation, we understand how our mentioning of ‘patient-specific representation’ in the abstract and discussion can be confusing. To avoid this confusion, we include these results in Sec. 6.6 of the modified version of the manuscript.
>
> **CONTRIBUTIONS**
> We modify the contributions section to better reflect what we believe the paper offers the broader machine learning community. To use your terminology, we compress our contributions to reflect a 'new angle' with with we approach the problem and 'best performance' that we achieve with CLOCS. 1) (New Angle) CLOCS, a family of patient-specific contrastive learning methods that exploit the spatial and temporal information present within electrocardiogram (ECG) signals. 2) (Best Performance) We show that CLOCS outperforms state-of-the-art contrastive learning methods such as SimCLR and BYOL, when evaluated on four diverse ECG datasets. Please refer to the modified version of the manuscript to see these changes.
>
> **TASK DESCRIPTION**
> The task we are solving for is cardiac arrhythmia classification which is of utmost importance in the medical domain given the potential consequences associated with a missed or improperly diagnosed cardiac abnormality.  To clarify how the time-series signals are associated with labels, we introduce Fig. 1, which also provides further intuition behind our proposed methods (CMSC and CMLC). Namely, each time-series signal of around 5 seconds in duration is associated with a single class label (related to a cardiac arrhythmia). To avoid confusing the ML community with medical jargon, we relegate the names of these arrhythmia to the Appendix (Appendix A).
>
> **MOTIVATION BEHIND CMSC AND CMLC**
> We modify the entirety of Section 4 to better address your concerns revolving around the intuition of the proposed methods (CMSC and CMLC in particular). More specifically, we introduce Fig. 1 which complements the text in Section 4 to illustrate both temporal and spatial invariances that are present within an ECG signal and which can be exploited for contrastive learning purposes. We hope this clears up any potential concerns.
>
> **STRONG CMSC PERFORMANCE AT E=128**
> As for the relatively strong generalization performance of CMSC at E=128, we direct the reviewer to Fig. 9 in Appendix G.1. Here, we show the distribution of the intra-patient and inter-patient distances as a function of the embedding dimension, E. We hypothesize that the following two observations explain the relative benefit of CMSC at E=128. First, we see that the intra-patient and inter-patient distributions are well separated (as opposed to those at E=32,64). This separation can be useful in distinguishing patients with different cardiac arrhythmia classes. However, E=256 also exhibits a high degree of distribution separation. Here, we introduce our second observation that of the results associated with E=128,256, those at E=128 exhibit lower intra-patient distances, on average. This implies that patient-specific representations are more tightly-clustered at E=128, a positive characteristic particularly when these representations map to the same class. Since our downstream task is one of classification, then such tightness is advantageous. We mention a concise version of this explanation in the modified version of the manuscript (Sec. 6.5, second paragraph).
>
> We hope the above responses and the modified version of the manuscript have addressed your concerns.

---

### Author Response · Authors · 2020-11-23
**Final Version of Manuscript**

We thank all the reviewers for the time and effort they have taken to read our manuscript and provide us with valuable feedback.

**FINAL MANUSCRIPT UPLOADED**
We have now finalized our manuscript and have uploaded a modified version of the main manuscript and the supplementary material. Here are the main changes, outlined at a high-level, that we have performed:

1) We have modified our **Contribution** statement to reflect the novel angle with which we approach contrastive learning and the state-of-the-art results that we achieve.
2) We have modified the **Related Work** section to remove our statement claiming that we are the first to perform contrastive learning on physiological signals.
3) We have modified the **Methods** section significantly to illustrate the temporal and spatial invariances within ECG signals that we exploit for contrastive learning purposes. We also provide further intuition underlying our proposed methods (e.g., CMSC, CMLC, CMSMLC).
4) We have improved our explanation of the patient-specific contrastive loss equation in **Sec. 4.3**.
5) We have included details about the duration of the ECG segments used during pre-training in **Sec. 5.2**.
6) We have conducted experiments using an ECG-specific pre-training method (which we call MT-SSL in the manuscript) and is based on the work in https://arxiv.org/abs/2002.03898. We find that CLOCS (our method) also outperforms MT-SSL. Please refer to **Sec. 6.1 and 6.2** for results. Implementation details for this method can be found in Appendix C.3.2.
7) We have included **Sec. 6.6** to provide some evidence that CLOCS leads to the learning of patient-specific representations.
8) To allow for others in the scientific community to reproduce our work, we have also made our anonymized code public at https://tinyurl.com/CLOCSSubmission.

We hope our responses to your reviews and the modified version of the manuscript have adequately addressed your concerns.

---

### Decision · Program_Chairs · 2021-01-07
**Final Decision**

**Decision:**

Reject

**Comment:**

This paper proposes a representation learning approach from cardiac signals, which adopts contrastive learning to incorporates knowledge on patient-specificity. This problem is highly motivating because of potential application to medicine and healthcare and large amounts of accumulating unlabeled physiological data. The presentation needs to be substantially improved – e.g., lack of clear description of the contribution, the details such as input data and network architecture, a clear description of the downstream tasks, missing explanations of equations, etc – some of which were addressed in the revision. Major concerns include lack of comparison with relevant prior methods developed for cardic signals, and the need for further refinement of domain knowledge-based intuitions.